# Chemical Composition and Histochemical Localization of Essential Oil from Wild and Cultivated *Rhaponticum carthamoides* Roots and Rhizomes

**DOI:** 10.3390/plants11152061

**Published:** 2022-08-06

**Authors:** Velislava Todorova, Stanislava Ivanova, Yoana Georgieva, Vanya Nalbantova, Diana Karcheva-Bahchevanska, Niko Benbassat, Martina S. Savova, Milen I. Georgiev, Kalin Ivanov

**Affiliations:** 1Department of Pharmacognosy and Pharmaceutical Chemistry, Faculty of Pharmacy, Medical University-Plovdiv, 4002 Plovdiv, Bulgaria; 2Department of Plant Cell Biotechnology, Center of Plant Systems Biology and Biotechnology, 4000 Plovdiv, Bulgaria; 3Laboratory of Metabolomics, Department of Biotechnology, The Stephan Angeloff Institute of Microbiology, Bulgarian Academy of Sciences, 139 Ruski Blvd, 4000 Plovdiv, Bulgaria

**Keywords:** *Rhaponticum carthamoides* (Willd.) Iljin, Maral root, secretory structures, GC-MS

## Abstract

*Rhaponticum carthamoides* (Willd.) Iljin is not only a source of phytosteroids and flavonoids, but is also source of essential oil (EO). This study evaluated the volatile metabolic constituents and histochemical localization of root and rhizome essential oils (EOs) from *R. carthamoides* populations wild-grown in Russia and cultivated in Bulgaria. The performed histochemical analysis confirmed the presence of lipophilic substances in the secretory ducts of the examined roots and rhizomes. Both EOs were obtained by hydrodistillation and further analyzed by gas chromatography with mass detection. The results showed differences between the chemical compositions of both EOs. Forty-six components were tentatively identified in *R. carthamoides* oil from the wild population, with *β*-selinene (4.77%), estragole (6.32%), D-carvone (6.37%), cyperene (8.78%), and ledene oxide (11.52%) being the major constituents. In the EO isolated from cultivated *R. carthamoides*, twenty-three compounds were tentatively identified, with humulene (7.68%), *β*-elemene (10.76%), humulene-1,2-epoxide (11.55%), ledene oxide (13.50%), and *δ-*elemene (19.08%) predominating. This is the first report describing the histolocalization and chemical profile of EO from *R. carthamoides* cultivated in Bulgaria. Further research on the cultivation of *R. carthamoides* in Bulgaria would affect the relationship between its chemical composition and pharmacological effects.

## 1. Introduction

*Rhaponticum carthamoides* (Willd.) Iljin (*R. carthamoides*) is a perennial plant that belongs to Asteraceae family and is also known as Maral root [1,2]. The species is native to South Siberia, the Altai and Sayan Mountains, where it grows at 1200–2300 m above sea level [1,3]. It has been used since ancient times in Russia, Mongolia, and China [1,2]. According to the Russian Pharmacopoeia, the usable parts of *R. carthamoides* are the roots and rhizomes, characterized by vertical, cylindrical, woody, branched, and irregularly wrinkled on the outside, 8–9 cm long and up to 3 cm thick [4]. The roots and rhizomes are source of phytoecdysteroids, which determine the anabolic, neuroprotective, hypoglycemic effects, and metabolism regulation [1,5,6,7,8,9]. *R. carthamoides* roots are source of flavonoids, which determine the antioxidative and hypolipidemic effects [1,5,10,11,12]. Moreover, *R. carthamoides* roots and rhizomes are also source of essential oil, and there are few reports describing the isolation and chemical characterization of the essential oil (EO) [1,5,13,14].

The chemical composition of EO is composed by monoterpene hydrocarbons, oxygenated monoterpenes, sesquiterpene hydrocarbons, and oxygenated sesquiterpenes [13,14]. The main reported compounds isolated from *R. carthamoides* EO are 13-norcypera-1(5),11(12)-diene (15.3–22.6%), aplotaxene (21.2%), cyperene (17.9–18.2%), cadalene (7.2%), geraniol, *α-*pinene, *β*-pinene, limonene, *β*-caryophyllene, and *β*-elemene. According to these studies, the EO showed antibacterial activity against *Enterococcus faecalis*, *Listeria monocytogenes*, *Staphylococcus aureus*, *Pseudomonas aeruginosa*, *Escherichia coli*, *Saccharomyces cerevisiae*, *Streptococcus pyogenes*, and *Candida albicans* with the minimum inhibitory concentration (MIC) ranging between 32 and 625 μg/mL. Moreover, the EO has been documented to possess antioxidant and anti-inflammatory activity [14].

The plant volatile oils and resins are synthesized and accumulated in specialized secretory structures, the type and localization of which are characterized mainly by the plant family and the species. The internal secretory structures, mostly termed as ducts or canals, are common in many species of the Asteraceae family and have also been reported in *R. carthamoides* [3,15]. The data about histochemical localization of EO in *R. carthamoides* roots and rhizomes are also limited [3]. Maral root is a non-native plant for the flora of Bulgaria, and there are no reported data about the isolation of EO from the plants cultivated in Bulgaria. In contrary, isolation, chemical composition and bioactivities were previously reported about *R. carthamoides* hairy roots and roots of soil-grown plants in other European countries [1,5,13,14,16].

The aim of the present study was to investigate and compare the chemical composition and histochemical localization of EOs in the roots and rhizomes of wild-grown and cultivated populations of *R. carthamoides*.

## 2. Materials and Methods

### 2.1. Plant Materials

Wild-grown *R. carthamoides* roots and rhizomes were purchased from Siberia (Russia), and cultivated *R. carthamoides* plants were purchased from Karlovo (Bulgaria). The cultivated plant has grown in a continental climate with an average annual temperature 11.9 °C and diluvial meadow cinnamon soil. The specimens were identified according to the Russian Pharmacopoeia [4].

### 2.2. Chemicals and Reagents

For determination of the retention indices (RI), the following hydrocarbons were used: octane (≥99%), nonane (99%), decane (≥99%), undecane (≥99%), dodecane (99%), tridecane (≥99%), tetradecane (≥99%), hexadecane (≥99%), heptadecane (99%), octadecane (99%), nonadecane (99%), eicosane (99%), heneicosane (≥99.5%), docosane (99%), tricosane (99%), tetracosane (99%), pentacosane (99%), hexacosane (99%), octacosane (99%), and triacontane (99%) purchased from Merck KGaA (Darmstadt, Germany). Hexane (GC grade) purchased from Thermo Fisher Scientific GmbH (Bremen, Germany) was used for dilution of the EOs.

### 2.3. Microscopic Histochemical Analysis

For the histochemical analysis, thin transverse sections (4–5 μm) of roots and rhizomes of both wild and cultivated *R. carthamoides* samples were obtained using a rotary microtome (Leica RM 2155) after the plant material had been fixed in a solution of 60% ethanol and glycerol in a ratio of 9:1, as a softening procedure of tissue prior to sectioning [17]. For the localization of lipophilic substances, the sections were treated with Sudan staining solution (Sudan III in 70% ethanol) for 20 min, rinsed in 50% ethanol to remove excess stain, and mounted in 50% glycerol [18]. Observations and photomicrographs were performed using a light microscope (Leica DM 2000 LED, Leica Microsystems, Wetzlar, Germany), equipped with a digital camera (Leica DMC 2900) and software for processing images (Leica Application Suite, LAS).

### 2.4. Isolation of Essential Oils

The EOs were isolated from dried roots and rhizomes by the hydrodistillation method. The dried roots and rhizomes of wild-grown and cultivated *R. carthamoides* (100 g) were chopped and subjected to hydrodistillation by the Clevenger-type apparatus for 4 h. The obtained EO samples were dried over anhydrous sodium sulphate and retained in brown vials in the refrigerator (4 °C) prior to use.

### 2.5. Chromatographic Condition

Both EOs were analyzed using gas chromatography with mass spectrometry (GC-MS). The GC-MS analyses were carried out using a Bruker Scion 436-GC SQ MS, Bremen, Germany. The ionization energy for the mass was 70 eV. The mass spectra were collected over the range of m/z 50–350 in full scan mode. The column was Bruker BR-5ms fused silica capillary (0.25 μm film thickness and 15 m × 0.25 mm i.d.). The injector was split/splitless with a split ratio 1:20. The oven temperature was initially held at 45 °C for 1 min and then increased to 160 °C at 5 °C/min and after that increased to 250 °C at 15 °C/min and then held for 1 min. Helium was the carrier gas with a flow rate of 1 mL/min. The temperatures of the detector and injector were 300 and 250 °C, respectively. The injection volume was 1 μL. The MS spectra of separated oil components were compared with their spectral data and retention indices with Wiley NIST11 Mass Spectral Library (NIST11/2011/EPA/NIH) and literature data. The RI values were calculated and compared to reported RI values for a C_8_-C_30_ series of *n*-alkane standards analyzed under the same GC conditions as above [19,20].

## 3. Results and Discussion

### 3.1. Histolocalization of Secretory Structures

Microscopic observations revealed the presence of internal secretory ducts filled with orange-brown secretion in the roots and rhizomes of the examined wild and cultivated samples of *R. carthamoides* (Figure 1 and Figure 2).

A characteristic feature in the root transverse sections of both investigated populations was the arrangement of the secretory ducts in the form of a ring along the endoderm between the parenchyma of the primary cortex and the secondary phloem (Figure 1A and Figure 2A).

In the rhizome transverse section of the wild *R. carthamoides*, the secretory ducts were scattered throughout the cortical parenchyma and the secondary phloem (Figure 1D), while in the rhizome of the cultivated population these oil secretory structures were observed as characteristic arches arranged on a group of lignified fibers, capping the vascular bundles (Figure 2D). Such distinctions can be attributed to the different stages of ontogenesis.

However, the secretory system structures in the examined *R. carthamoides* roots and rhizomes were similar. The secretory ducts were consisted of intercellular spaces surrounded by specialized epithelial cells responsible for the volatile oil secretion (Figure 1B,E and Figure 2B,E).

The lipid nature of the duct content and related surrounding epithelial cells was confirmed by orange-red staining with Sudan III (Figure 1C,F and Figure 2C,F). No differences were detected between wild and cultivated samples of the *R. carthamoides* root and rhizome from a histochemical perspective.

The results of the performed microscopic histochemical analysis were in accordance with the data specified in the Russian Pharmacopoeia [4] and also corresponded to the anatomical studies on the vegetative organs of *R. carthamoides* [3].

### 3.2. Volatile Constituents in Rhaponticum carthamoides Roots and Rhizomes Essential Oil

The extracted EOs were analyzed by means of GC-MS. The color of the resulting EO from the wild-growing plant was yellow, and from the cultivated it was dark yellow. The odor of both EOs was exotic.

Chemical analysis resulted in identification of 46 volatile compounds in the EO from wild-grown *R. carthamoides*, representing 76.46% of the total oil, while 23 volatile compounds were identified in the oil from the plants cultivated in Bulgaria, which amounted to 83.07% of the total oil. In Figure 3 and Figure 4 are presented chromatograms of *R. carthamoides* EO from wild and cultivated samples, respectively.

A comparative GC-MS analysis of EOs from *R. carthamoides* wild and cultivated populations showed that the chemical compounds of both EOs were different. Table 1 presents the chemical composition of EOs with their formulas, retention indices, and relative percentage amounts.

The *R. carthamoides* EO of wild-grown plants appeared rich in volatile compounds from monoterpene hydrocarbons (1.31%), oxygenated monoterpenes (20.90%), sesquiterpene hydrocarbons (28.23%), and oxygenated sesquiterpenes (17.07%).

Monoterpene hydrocarbons detected from *R. carthamoides* EO from the wild population were *α*-pinene, *p*-cymene, D-limonene, and *p*-cymenene, and their percentage of the total was 1.31%. The main oxygenated monoterpene compounds were estragole (6.32%) and D-carvone (6.37%); the rest of the oxygenated monoterpenes were *p*-menth-1-en-9-al, *β*-linalool, thujone, *cis*-pinocarveol, borneol, (-)-camphor, terpinen-4-ol, *p*-cymen-8-ol, *cis*-carveol, α-terpineol, *p*-menth-1-en-3-one, anethole, thymol, and carvacrol, and their percentages were less than 2%. Sesquiterpene hydrocarbons were the major fractions in both oils. Sesquiterpene hydrocarbons in *R. carthamoides* EO from a wild population were 28.23%, and the main chemical compounds were cyperene (8.78%), *δ*-elemene (2.35%), *β*-elemene (2.35%), *β*-selinene (4.77%), and *β*-guaiene (3.71%). The rest of the sesquiterpenes were less than 2% (*α*-copaene, *α*-gurjunene, methyleugenol, *α*-bergamotene, humulene, rotundene, *γ*-elemene, *cis*-thujopsene, *α*-bulnesene, aromandrene, and *γ*-cadinene). The fraction of oxygenated sesquiterpenes in *R. carthamoides* EO from the wild population was 8.95%. Ledene oxide represented 11.52% of the oxygenated sesquiterpenes fraction, and the remaining compounds were (+)-spathulenol, caryophyllene oxide, *β*-eudesmol, humulene-1,6-dien-3-ol, allo aromadendrene oxide, and isoaromandrene epoxide.

The EO from cultivated *R. carthamoides* appeared rich in volatile compounds classified as sesquiterpene hydrocarbons (45.10%) and oxygenated sesquiterpenes (34.52%).

Monoterpene hydrocarbons and oxygenated monoterpenes were not found in the EO from the cultivated sample. Sesquiterpene hydrocarbons (45.10%) were the main fraction in *R. carthamoides* EO from a cultivated population. The main sesquiterpene hydrocarbons were humulene (7.68%), *β*-elemene (10.76%), and *δ*-elemene (19.08%). The remaining sesquiterpene hydrocarbons made up less than 2% (longifolene, *β*-caryophyllene, *γ*-elemene, *cis*-thujopsene, *α*-selinene, *β*-selinene, and *β*-bisabolene). Oxygenated sesquiterpenes comprised 34.52% of the total oil. The main isolated oxygenated sesquiterpenes were ledene oxide (13.50%), humulene-1,2-epoxide (11.55%), caryophyllene oxide (3.13%), and isoshyobunone (4.38%). Lower-amount oxygenated sesquiterpenes were bergamotol, *β*-santalol, cubenol, and bisabolene epoxide.

It is worth noting that the hydrocarbon and oxygenated monoterpene fractions appeared only in *R. carthamoides* EO from the wild population. Moreover, some similarities in chemical composition between oil samples were observed. The sesquiterpene hydrocarbons and oxygenated sesquiterpene fractions predominated in both oils. Compounds common to both EOs were *δ*-elemene, *β*-elemene, *α*-bergamotene, *β*-selinene, caryophyllene oxide, and ledene oxide, but the concentration in the oil from the wild population was lower. The hydrocarbon sesquiterpene *β*-guaiene was found in the wild but not in the cultivated population. On the other hand, the oxygenated sesquiterpenes humulene-1,2-epoxide, isoshyobunone were found only in the cultivated population.

The major fraction of sesquiterpene hydrocarbons in *R. carthamoides* EO from the wild and cultivated populations were also previously reported in EOs from *R. carthamoides* hairy roots and roots of soil-grown plant [15]. On the contrary, in EO isolated from *R. carthamoides* roots and rhizomes grown in Poland, monoterpenes were the prevalating fraction [13]. The following volatile compounds—13-norcypera-1(5),11(12-diene) (15.3 and 22.6%), cyperene (17.9% and 18.2%), and cadalene (7.2%)—had previously been reported as the main chemical compounds isolated from *R. carthamoides* hairy root EO [13,14]. We found that cyperene was one of the major compounds in *R. carthamoides* EO from the wild population, but this compound was not found in EO from the cultivated population. Humulene was not previously reported, but we found it in both oils, and in the cultivated sample concentration the percentage was higher. Previous studies reported *β*-elemene in a lower concentration than was reported in the current study. Although *β*-elemene was identified in both samples, this compound was found in a higher amount in the oil isolated from the cultivated population. The concentration of *β*-selinene was almost the same when comparing the amounts previously reported and those discovered in the current investigation. Caryophyllene oxide had been found previously, and it was found in both samples in our study. Alpha-pinene, rotundene, and limonene had been found in previous studies [13,14], and our performed analyses confirmed their presence only in the EO from the wild population. The following major compounds: cyperene, *trans*-*β*-bergamotene, petasitene, 2,5,8-trimethyl-1-naphthol, dauca-4(11),8-diene, nardosina-1(10),11-diene, and aplotaxene were found in the previously reported chemical composition of EO from *R. carthamoides* roots of soil-grown plants. In contract, in both of our analyzed Eos, cyperene was found only in the sample from the wild-grown *R. carthamoides*. The major compounds in *R. carhamoides* EO from the wild population estragole, D-carvone, *β*-selinene, and ledene oxide were not previously reported in EO from *R. carthamoides* roots of soil-grown plants. Moreover, among the major chemical compounds in EO from the cultivated population, only humulene was previously reported in the oil sample from *R. carthamoides* roots of soil-grown plants [14].

The results obtained in this study showed differences in chemical composition of both analyzed EOs. The contents of compounds and their amounts in EOs may vary depending on geographic origin, individual plant chemotypes, time of harvest, and habitat conditions—soil, soil management, and climate [21,22,23]. Different chemical compositions of EOs may lead to changes in the biological activity. In this regard, strict monitoring of chemical composition is important if different sources of plant material are used. Further studies are needed to investigate the potentially altered pharmacological properties of the EO from *R. carthamoides* populations and the relationship between chemical composition and pharmacological effects. Moreover, further studies on the cultivation conditions of *R. carthamoides* in Bulgaria are necessary. This may increase the importance and use of *R. carthamoides* EO.

## 4. Conclusions

The histochemical analysis of *R. carthamoides* roots and rhizomes from wild and cultivated populations confirmed the presence of lipophilic substances in the secretory ducts. The conducted histochemical analysis could assist the pharmacognostic one and in particular the microscopic identification of the tested herbal drug, which could improve the quality control in the presence of adulteration. In addition, it gives information on the exact localization of the contained EO as a secondary metabolite. Furthermore, the data obtained from GC-MS analysis indicated that EO from the *R. carthamoides* cultivated population did not contain monoterpene hydrocarbons. In EOs from the wild and cultivated populations, the sesquiterpene hydrocarbons were the major fraction at 28.23% and 45.10%, respectively. The major compounds in EO from the wild population were *β*-selinene (4.77%), estragole (6.32%), D-carvone (6.37%), cyperene (8.78%), and ledene oxide (11.52%). The main identified compounds in the cultivated population were humulene (7.68%), *β*-elemene (10.76%), humulene-1,2-epoxide (11.55%), ledene oxide (13.50%), and *δ*-elemene (19.08%).

The knowledge of *R. carthamoides* cultivated in Bulgaria is enhanced by investigating for the first time the composition of its EO. In view of our results regarding the differences in chemical composition between the EOs from the wild and cultivated populations, further studies on improving the cultivation conditions of *R. carthamoides* in Bulgaria may be necessary.

## Figures and Tables

**Figure 1 plants-11-02061-f001:**
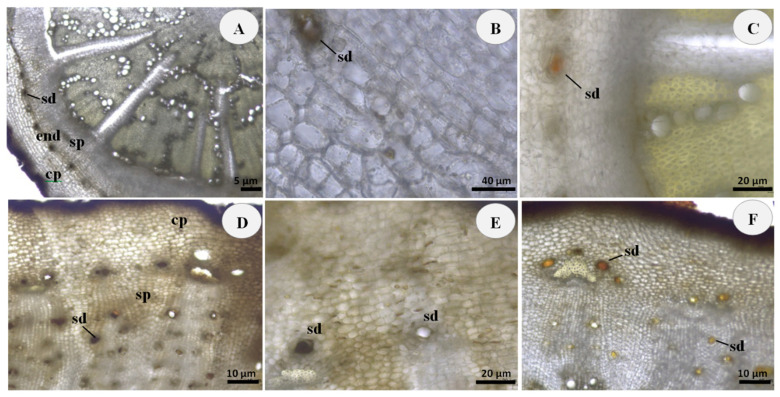
Root and rhizome transverse sections of wild *R. carthamoides*: (**A**) General aspect of unstained root section, showing the presence of secretory ducts (sd) arranged in a ring along the endoderm (end) between the cortical parenchyma (cp) and secondary phloem (sp); (**B**) Details of root secretory ducts (sd) filled with orange-brown secretion; (**C**) Secretory ducts (sd) of root stained orange-red with Sudan III for the presence of lipophilic substances; (**D**) Unstained rhizome section, showing the presence of secretory ducts (sd) located in the cortical parenchyma (cp) and the secondary phloem (sp); (**E**) Details of rhizome secretory ducts (sd) filled with orange-brown secretion; (**F**) Secretory ducts (sd) of rhizome stained orange-red with Sudan III for the presence of lipophilic substances. Scale bars: (**A**) = 5 μm; (**B**) = 40 μm; (**C**,**E**) = 20 μm; (**D**,**F**) = 10 μm.

**Figure 2 plants-11-02061-f002:**
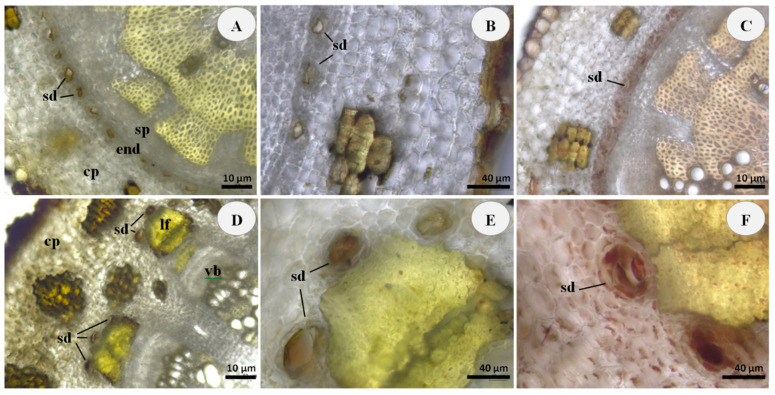
Root and rhizome transverse sections of cultivated *R. carthamoides*: (**A**) General aspect of unstained root section, showing the presence of secretory ducts (sd) arranged in a ring along the endoderm (end) between the cortical parenchyma (cp) and secondary phloem (sp); (**B**) Details of root secretory ducts (sd), containing orange-brown secretion; (**C**) Secretory ducts (sd) of root stained orange-red with Sudan III for the presence of lipophilic substances; (**D**) Unstained rhizome section, showing the presence of secretory ducts (sd) located on the group of lignified fibers (lf) capping the vascular bundles (vb); (**E**) Details of rhizome secretory ducts (sd) filled with orange-brown secretion; (**F**) Secretory ducts (sd) of rhizome stained orange-red with Sudan III for the presence of lipophilic substances; Scale bars: (**A**,**C**,**D**) = 10 μm; (**B**,**E**,**F**) = 40 μm.

**Figure 3 plants-11-02061-f003:**
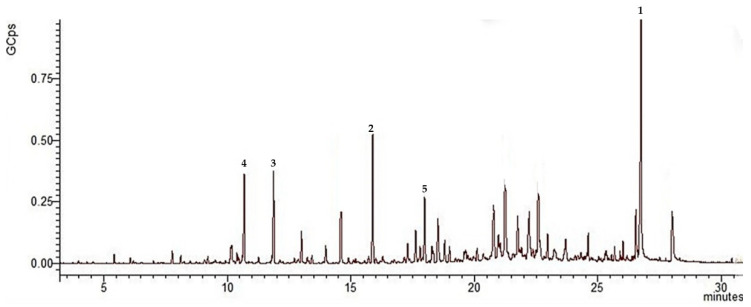
Chromatogram of *R. carthamoides* EO from the wild population, compounds derived from GC-MS analysis, where GCps—Giga Counts per second and the numbers refer to the following: 1—ledene oxide, 2—cyperene, 3—D-carvone, 4—estragole, and 5—*β*-selinene.

**Figure 4 plants-11-02061-f004:**
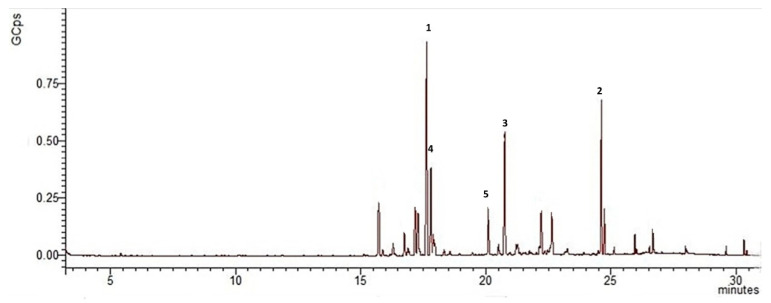
Chromatogram of *R. carthamoides* EO cultivated in Bulgaria, compounds derived from GC-MS analysis, where GCps—Giga Counts per second and the numbers refer to the following: 1—δ-elemene, 2—ledene oxide, 3—humulene-1,2-epoxide, 4—β-elemene, and 5—humulene.

**Table 1 plants-11-02061-t001:** Volatile organic composition of *R. carthamoides* isolated EO from wild-grown population (WP) and cultivated in Bulgaria population (CP) as a percentage of total EO, tr—trace (less than 0.05%), “-”—not detected compounds. The percentage distribution of the main terpene classes identified in the analyzed EOs is summarized at the end of the table.

No	Compound	RI	Formula	Class of Compound	% of Total in WP	% of Total in CP
1	*α*-Pinene	924	C_10_H_16_	MH	0.15	-
2	*p*-Cymene	1007	C_10_H_14_	MH	0.3	-
3	Limonene	1012	C_10_H_16_	MH	0.10	-
4	*p*-Menth-1-en-9-al	1057	C_10_H_16_O	MO	tr	-
5	*p*-Cymenene	1074	C_10_H_14_	MH	0.76	-
6	*β*-Linalool	1088	C_10_H_18_O	MO	0.42	-
7	Thujone	1102	C_10_H_16_O	MO	0.10	-
8	*cis*-Pinocarveol	1129	C_10_H_16_O	MO	tr	-
9	(-)-Camphor	1130	C_10_H_16_O	MO	0.40	-
10	Borneol	1159	C_10_H_18_O	MO	0.10	-
11	Terpinen-4-ol	1169	C_10_H_18_O	MO	1.94	-
12	*p*-Cymen-8-ol	1177	C_10_H_14_O	MO	1.02	-
13	*α*-Terpineol	1186	C_10_H_18_O	MO	0.41	-
14	Estragole	1189	C_10_H_12_O	MO	6.32	-
15	*cis*-Carveol	1211	C_10_H_16_O	MO	tr	-
16	D-carvone	1236	C_10_H_14_O	MO	6.37	-
17	*p*-Menth-1-en-3-one	1247	C_10_H_18_O	MO	tr	-
18	Anethole	1280	C_10_H_12_O	MO	1.93	-
19	Thymol	1288	C_10_H_14_O	MO	0.10	-
20	Carvacrol	1297	C_10_H_14_O	MO	1.79	-
21	*α*-Copaene	1369	C_15_H_24_	SH	tr	-
22	*δ*-Elemene	1380	C_15_H_24_	SH	2.35	19.08
23	*β*-Elemene	1389	C_15_H_24_	SH	2.35	10.76
24	Longifolene	1390	C_15_H_24_	SH	-	0.10
25	Cyperene	1394	C_15_H_24_	SH	8.78	-
26	*α*-Gurjunene	1398	C_15_H_24_	SH	0.10	-
27	Methyleugenol	1399	C_11_H_14_O_2_	O	tr	-
28	*β*-Caryophyllene	1413	C_15_H_24_	SH	-	1.16
29	*α*-Bergamotene	1428	C_15_H_24_	SH	0.10	2.29
30	*γ*-Elemene	1430	C_15_H_24_	SH	1.23	0.38
31	Humulene	1448	C_15_H_24_	SH	0.10	7.68
32	Aromandrene	1450	C_15_H_24_	SH	0.63	-
33	Rotundene	1459	C_15_H_24_	SH	1.29	-
34	*cis*-Thujopsene	1473	C_15_H_24_	SH	0.36	1.83
35	*α*-Selinene	1475	C_15_H_24_	SH	-	1.23
36	*β*-Selinene	1476	C_15_H_24_	SH	4.77	0.49
37	*α*-Bulnesene	1490	C_15_H_24_	SH	1.19	-
38	*β*-Guaiene	1497	C_15_H_24_	SH	3.71	-
39	*β*-Bisabolene	1502	C_15_H_24_	SH	-	0.10
40	Myristicine	1508	C_11_H_12_O_3_	O	1.40	-
41	*γ*-Cadinene	1532	C_15_H_24_	SH	1.27	-
42	(+)-Spathulenol	1556	C_15_H_24_O	SO	1.29	-
43	Caryophyllene oxide	1560	C_15_H_24_O	SO	0.84	3.13
44	Humulene-1,2-epoxide	1592	C_15_H_24_O	SO	-	11.55
45	*β*-Eudesmol	1604	C_15_H_26_O	SO	0.37	-
46	Isoaromandrene epoxide	1620	C_15_H_24_O	SO	0.66	-
47	Humulene-1,6-dien-3-ol	1621	C_15_H_26_O	SO	0.97	-
48	Apiole	1640	C_12_H_14_O_4_	O	1.37	-
49	Isoshyobunone	1650	C_15_H_24_O	SO	-	4.38
50	Allo aromadendrene oxide	1660	C_15_H_24_O	SO	1.42	-
51	*β*-Santalol	1670	C_15_H_24_O	SO	-	0.21
52	Cubenol	1672	C_15_H_26_O	SO	-	1.06
53	Bergamotol	1679	C_15_H_24_O	SO	-	0.10
54	Bisabolene epoxide	1680	C_15_H_24_O	SO	-	0.59
55	Ledene oxide	1682	C_15_H_24_O	SO	11.52	13.50
56	Palmitic acid	1890	C_16_H_32_O_2_	O	-	1.79
57	Linoleic acid	2110	C_18_H_32_O_2_	O	6.18	0.23
58	Arachidonic acid	2224	C_20_H_32_O_2_	O	-	tr
59	11, 14, 17-Eicosatrienoic acid, methyl ester	2240	C_20_H_34_O_2_	O	-	4.84
	Terpene classes					
	Monoterpene hydrocarbons (MH)				1.31	-
	Oxygenated monoterpenes (MO)				20.90	-
	Sesquiterpene hydrocarbons (SH)				28.23	45.10
	Oxygenated sesquiterpenes (SO)				17.07	34.52
	Others (O)				8.95	6.86
	Total identified				76.46	86.48

## Data Availability

Not applicable.

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
