# Peer review of "Chemical Composition and Histochemical Localization of Essential Oil from Wild and Cultivated Rhaponticum carthamoides Roots and Rhizomes"

_plants, 2022, doi:10.3390/plants11152061_

Round 1
Reviewer 1 Report
Journal: Plants
Manuscript: Manuscript ID: plants-1825766
Title: Chemical Composition and Histochemical Localization of Essential Oil from Wild and Cultivated Rhaponticum carthamoides Roots and Rhizomes
Dear Editor
Thank you for inviting me to review the manuscript “Chemical Composition and Histochemical Localization of Essential Oil from Wild and Cultivated Rhaponticum carthamoides Roots and Rhizomes”. The publication requires addition of information and reediting; also, the photographs should have better quality.
Kind regards,
Abstract
1. The abstract should include the following elements: place the introduction addressed in a broad context, highlight the purpose of the study, describe briefly the main methods applied, summarize the main findings, and indicate the main conclusions.
Keywords
2. Eliminate phrases appearing in the title of the manuscript.
Introduction
3. l. 35-46 this is general information; I suggest adding relevant data related to the topic of the study
4. l. 37 – 39 - Describe the activity with reference to the presented species (for antibacterial activity, add information about the MIC value and the specific bacterial strain)
5. l. 49 when mentioned the first time, “Rhaponticum carthamoides (Willd.) Iljin” should be used; hereinafter use “R. carthamoides”. In this fragment, delete “R. carthamoides”.
6. l. 52 – 54 a. Specify which extract precisely is meant here
b. provide detailed information from the cited publications for the individual biological properties
c. provide the MIC value and specific bacterial strains for the antimicrobial activity
7. l. 54 It should be “[9-12] instead of “[9,10,11,12].”
8. l. 54 – 58 combine the general information about the morphology of the organ with relevant knowledge of e.g. phytoecdysteroids (ecdysones, phytoecdysones) and other biologically active chemical compounds with phytotherapeutic activity etc. using the latest literature reports on Rhaponticum carthamoides
9. l. 59-60 while citing 7 literature references, provide substantive information associated with to the topic of the study, eliminate listing secondary metabolites “R. carthamoides is a source of phytosteroids, flavonoids, phenolic acids and EO phytosteroids, flavonoids, phenolic acids and EO”; instead provide specific information related to the subject of the study
10. l. 60-61. Provide information about these reports in connection with the topic of the study
“There are few reports describing the isolation and chemical characterization of EO from R. carthamoides roots [9,14,15].”
11. l. 63-64 please, see comment 4
12. l.65-68 Strengthen the rationale of the research skillfully connecting it with what has been done and what is missing in the literature. I am providing several references, please also search for others.
1. Geszprych, A., & Weglarz, Z. (2002). Composition of essential oil from underground and aboveground organs of Rhaponticum carthamoides [Willd.] Iljin. Herba polonica, 4(48).
2. Skała, E., Rijo, P., Garcia, C., Sitarek, P., Kalemba, D., Toma, M., ... & Śliwiński, T. (2016). The essential oils of Rhaponticum carthamoides hairy roots and roots of soil-grown plants: chemical composition and antimicrobial, anti-inflammatory, and antioxidant activities. Oxidative medicine and cellular longevity, 2016.
3. Todorova, V., Ivanov, K., & Ivanova, S. (2021). Comparison between the Biological Active Compounds in Plants with Adaptogenic Properties (Rhaponticum carthamoides, Lepidium meyenii, Eleutherococcus senticosus and Panax ginseng). Plants, 11(1), 64.
13. l. 44-48, l. 65-68, l. 83-85 one or two sentences cannot form a paragraph, especially a subsection (please check this throughout the manuscript)
Materials and Methods
14. l. 113 – 173 change the order of the subsections keeping the chronological arrangement. “Plant Materials” should be the first subsection
15. l. 83 – 85. a. Give the geographical coordinates of the cultivation area of the plants “from the region of Karlovo”
b. At what stage were the plants collected for the analyses?
16. l. 86-93. Add the following information:
a. what plant material was cut (fresh or fixed)?
b. what thickness were the sections?
c. what were the sections cut with?
d. provide a description of the histological assays used
e. eliminate the mental shortcuts “according to standard procedures”
f. complete literature references for the methods used
Results and Discussion
17. Check the guidelines for authors for the “Results and Discussion”
18. Figure 1A-E, 2A-F: the photographs are poor quality (this may be related to the large thickness of the hand-cut sections), please provide photographs with good graphic quality
19. The bars are not visible; some photographs have no bars at all
20. in subsection 2.3, complete the discussion: the two references cited are not a proper discussion
21. Latin names should be italicized (figure 1-4 and throughout the manuscript)
22. Table 1. Explain all abbreviations. What does the symbol “_” stand for???,
23. l. 225 - 239 do not cite the same references [14, 15] five times – once is enough
24. In subsection 3.2, expand the discussion; discuss the obtained results step by step with the current literature reports from other authors
25. Please specify for which group of readers the presented research will be useful
26. Indicate the research perspectives in the future
Conclusion
27. l. 249-253 this information should not be placed here
28 L. 263-265 please re-think the sentence, especially “…the diagnostic purposes”... ??, „…quality control of the plants’ drugs.” ???
29. Draw logical and concrete conclusions from the present study
References
30. Check the list of references (l. 307-309, 317-322 etc.) and the entire text so that they follow the guidelines for authors
Author Response
Dear reviewer,
Thank you for your remarks and recommendations.
Reviewer 1 comments:
- Reviewer 1 comment 1:
Abstract
- The abstract should include the following elements: place the introduction addressed in a broad context, highlight the purpose of the study, describe briefly the main methods applied, summarize the main findings, and indicate the main conclusions.
Authors’ response:
Dear reviewer,
Thank you for your recommendations. We have reedited the abstract according to the requirements. The new abstract is as follow:
Rhaponticum carthamoides (Willd.) Iljin is not only a source of phytosteroids and flavonoids, but is also source of essential oil (EO). This study evaluated the volatile metabolic constituents and histochemical localization of root and rhizome essential oils (EOs) from Rhaponticum carthamoides (Willd.) Iljin populations wild-grown in Russia and cultivated in Bulgaria. The performed histochemical analysis confirmed the presence of lipophilic substances in the secretory ducts of the examined roots and rhizomes. Both EOs were obtained by hydrodistillation and further analyzed by gas chromatography with mass detection. The results showed differences between the chemical compositions of the both EOs. Forty-six components were tentatively identified in R. carthamoides oil from the wild population, with b-selinene (4.77%), estragole (6.32%), D-carvone (6.37%), cyperene (8.78%) and ledene oxide (11.52%) being the major constituents. In the EO isolated from cultivated R. carthamoides, twenty-three compounds were tentatively identified, with humulene (7.68%), b-elemene (10.76%), humulene-1,2-epoxide (11.55%), ledene oxide (13.50%) and d-elemene (19.08%) predominating. This is the first report describing the histolocalization and chemical profile of EO from R.carthamoides cultivated in Bulgaria. Further research on the cultivation of R. carthamoides in Bulgaria would affect the relationship between its chemical composition and pharmacological effects.
- Reviewer 1 comment 2:
Keywords
- Eliminate phrases appearing in the title of the manuscript.
Authors’ response:
Dear reviewer,
Thank you for your recommendations. We have changes the keywords as follow: Rhaponticum carthamoides (Willd.) Iljin; Maral root; secretory structures; GC-MS
- Reviewer 1 comments 3 and 4:
Introduction
- l. 35-46 this is general information; I suggest adding relevant data related to the topic of the study
- l. 37 – 39 - Describe the activity with reference to the presented species (for antibacterial activity, add information about the MIC value and the specific bacterial strain)
Authors’ response:
Dear reviewer,
Thank you for your recommendations. We have deleted this paragraph.
- Reviewer 1 comment 5:
- l. 49 when mentioned the first time, “Rhaponticum carthamoides(Willd.) Iljin” should be used; hereinafter use “R. carthamoides”. In this fragment, delete “R. carthamoides”.
Authors’ response:
Dear reviewer,
Thank you for your recommendations. We have corrected it.
- Reviewer 1 comment 6:
- l. 52 – 54 a. Specify which extract precisely is meant here
- provide detailed information from the cited publications for the individual biological properties
- provide the MIC value and specific bacterial strains for the antimicrobial activity
Authors’ response:
Dear reviewer,
Thank you for your recommendations. We have deleted this paragraph, because it was off topic.
- Reviewer 1 comment 7:
- l. 54 It should be “[9-12] instead of “[9,10,11,12].”
Dear reviewer,
Thank you for your recommendations. We have corrected it.
- Reviewer 1 comment 8:
- l. 54 – 58 combine the general information about the morphology of the organ with relevant knowledge of e.g. phytoecdysteroids (ecdysones, phytoecdysones) and other biologically active chemical compounds with phytotherapeutic activity etc. using the latest literature reports on Rhaponticum carthamoides
Dear reviewer,
Thank you for your recommendations. We have edited it according your recommendations.
According to the Russian Pharmacopoeia, the usable parts of R. carthamoides are roots and rhizomes, characterized by vertical, cylindrical, woody, branched, and irregularly wrinkled on the outside, 8-9 cm long and up to 3 cm thick [4]. The roots and rhizomes are source of phytoecdysteroids, which determine the anabolic, neuroprotective, hypoglycemic effects, and metabolism regulation [1,5-9]. R. carthamoides roots are source of flavonoids which determine the antioxidative and hypolipidemic effects [1,5,10-12]. Moreover, R. carthamoides roots and rhizomes are also source of essential oil and there are few reports describing the isolation and chemical characterization of the EO [1,5,13,14].
- Reviewer 1 comment 9:
- l. 59-60 while citing 7 literature references, provide substantive information associated with to the topic of the study, eliminate listing secondary metabolites “R. carthamoides is a source of phytosteroids, flavonoids, phenolic acids and EO phytosteroids, flavonoids, phenolic acids and EO”; instead provide specific information related to the subject of the study
Authors’response:
Dear reviewer,
Thank you for your recommendations. We have edited it according your recommendations.
- Reviewer 1 comment 10:
- l. 60-61. Provide information about these reports in connection with the topic of the study
“There are few reports describing the isolation and chemical characterization of EO from R. carthamoides roots [9,14,15].”
Authors’ response:
Dear reviewer,
Thank you for your recommendations. We have corrected it.
- Reviewer 1 comment 10:
- l. 63-64 please, see comment 4
Authors’ response:
Dear reviewer,
Thank you for your recommendations. We include the following data:
The chemical composition of EO is composed by monoterpene hydrocarbons, oxygenated monoterpenes, sesquiterpene hydrocarbons and oxygenated sesquiterpenes [13,14]. The main reported compounds isolated from R. carthamoides EO are 13-norcypera-1(5),11(12)-diene (15.3-22.6%), aplotaxene (21.2%), cyperene (17.9-18.2%), cadalene (7.2%), geraniol, a-pinene, b-pinene, limonene, b-caryophyllene and b-elemene. According to these researches the EO showed antibacterial activity against Enterococcus faecalis, Listeria monocytogenes, Staphylococcus aureus, Pseudomonas aeruginosa, Escherichia coli, Saccharomyces cerevisiae and Streptococcus pyogenes, Candida albicans with the minimum inhibitory concentration (MIC) ranged between 32 and 625 mg/mL. Moreover, the EO has been documented to possess antioxidant and anti-inflammatory activity [14].
- Reviewer 1 comment 10:
- l.65-68 Strengthen the rationale of the research skillfully connecting it with what has been done and what is missing in the literature. I am providing several references, please also search for others.
- Geszprych, A., & Weglarz, Z. (2002). Composition of essential oil from underground and aboveground organs of Rhaponticum carthamoides [Willd.] Iljin. Herba polonica, 4(48).
- Skała, E., Rijo, P., Garcia, C., Sitarek, P., Kalemba, D., Toma, M., ... & Śliwiński, T. (2016). The essential oils of Rhaponticum carthamoides hairy roots and roots of soil-grown plants: chemical composition and antimicrobial, anti-inflammatory, and antioxidant activities. Oxidative medicine and cellular longevity, 2016.
- Todorova, V., Ivanov, K., & Ivanova, S. (2021). Comparison between the Biological Active Compounds in Plants with Adaptogenic Properties (Rhaponticum carthamoides, Lepidium meyenii, Eleutherococcus senticosus and Panax ginseng). Plants, 11(1), 64.
Authors’ response:
Dear reviewer,
Thank you for your recommendations. We have edited it.
- Reviewer 1 comment 13:
- l. 44-48, l. 65-68, l. 83-85 one or two sentences cannot form a paragraph, especially a subsection (please check this throughout the manuscript)
Authors’ response:
Dear reviewer,
Thank you for your remark. We have change it.
- Reviewer 1 comment 14:
Materials and Methods
- l. 113 – 173 change the order of the subsections keeping the chronological arrangement. “Plant Materials” should be the first subsection
Authors’ response:
Dear reviewer,
Thank you for your recommendations. We have changed it.
- Reviewer 1 comment 15:
- l. 83 – 85. a. Give the geographical coordinates of the cultivation area of the plants “from the region of Karlovo”
- At what stage were the plants collected for the analyses?
Authors’ response:
Dear reviewer,
Thank you for your recommendations. We have purchased it. It’s added in the text.
- Reviewer 1 comment 16:
- l. 86-93. Add the following information:
- what plant material was cut (fresh or fixed)?
- what thickness were the sections?
- what were the sections cut with?
- provide a description of the histological assays used
- eliminate the mental shortcuts “according to standard procedures”
- complete literature references for the methods used
Authors’ response:
Dear reviewer,
Thank you for your recommendations. The information had been added.
For the histochemical analysis, thin transverse sections (4-5 μm) of roots and rhizomes of both wild and cultivated R. carthamoides samples were obtained using a rotary microtome (Leica RM 2155) after the plant material had been fixed in a solution of 60% ethanol and glycerol in a ratio of 9:1, as a softening procedure of tissue prior to sectioning [17]. For the localization of lipophilic substances the sections were treated with Sudan staining solution (Sudan III in 70 % ethanol ) for 20 min, rinsed in 50% ethanol to remove excess stain and mounted in 50% glycerol [18]. Observations and photomicrographs were performed using a light microscope (Leica DM 2000 LED, Leica Microsystems, Germany), equipped with digital camera (Leica DMC 2900) and software for processing images (Leica Application Suite, LAS).
- Reviewer 1 comment 17:
Results and Discussion
- Check the guidelines for authors for the “Results and Discussion”
Authors’ response:
Dear reviewer,
Thank you for your recommendations. We have read it.
- Reviewer 1 comment 18:
- Figure 1A-E, 2A-F: the photographs are poor quality (this may be related to the large thickness of the hand-cut sections), please provide photographs with good graphic quality
Authors’ response:
Dear reviewer,
Thank you for your recommendations. Since there are 6 photos, the description should be as one file, not photo by photo, so we have combined the 6 photos into one file, which probably worsened the quality of the photos. The final resolution of the figures is as described in requirements of the journal.
- Reviewer 1 comment 19:
- The bars are not visible; some photographs have no bars at all
Authors’ response:
Dear reviewer,
Thank you for your recommendations. We have increased the scale bar on the figures and added them individually for each picture to the legend.
- Reviewer 1 comment 20:
- in subsection 2.3, complete the discussion: the two references cited are not a proper discussion
Authors’ response:
Dear reviewer,
Thank you for your recommendations. We edited it.
- Reviewer 1 comment 21:
- Latin names should be italicized (figure 1-4 and throughout the manuscript)
It’s done
Authors’ response:
Dear reviewer,
Thank you for your recommendations. We have changed it.
- Reviewer 1 comment 22:
- Table 1. Explain all abbreviations. What does the symbol “_” stand for???,
Authors’ response:
Dear reviewer,
Thank you for your recommendations. We added that “-“ means not detected.
- Reviewer 1 comment 23:
- l. 225 - 239 do not cite the same references [14, 15] five times – once is enough
Authors’ response:
Dear reviewer,
Thank you for your recommendations. We have corrected it.
- Reviewer 1 comment 24:
- In subsection 3.2, expand the discussion; discuss the obtained results step by step with the current literature reports from other authors
Authors’ response:
Dear reviewer,
Thank you for your recommendations. We have edited it.
The EO from cultivated R. carthamoides appeared rich in volatile compounds classified as sesquiterpene hydrocarbons (45.10%) and oxygenated sesquiterpenes (34.52%). Monoterpene hydrocarbons and oxygenated monoterpenes were not found in the EO from the cultivated sample. Sesquiterpene hydrocarbons (45.10%) were the main fraction in R. carthamoides EO from a cultivated population. The main sesquiterpene hydrocarbons were humulene (7.68%), b-elemene (10.76%) and d-elemene (19.08%). The remaining sesquiterpene hydrocarbons made up less than 2% (longifolene, b-caryophyllene, g-elemene, cis-thujopsene, a-selinene, b-selinene and b-bisabolene). Oxygenated sesquiterpenes comprised 34.52% of the total oil. The main isolated oxygenated sesquiterpenes were ledene oxide (13.50%), humulene-1,2-epoxide (11.55%), caryophyllene oxide (3.13%) and isoshyobunone (4.38%). Lower-amount oxygenated sesquiterpenes were bergamotol, b-santalol, cubenol and bisabolene epoxide.
It is worth noting that the hydrocarbon and oxygenated monoterpene fractions appeared only in R. carthamoides EO from the wild population. Moreover, some similarities in chemical composition between oil samples were observed. The sesquiterpene hydrocarbons and oxygenated sesquiterpene fractions predominated in both oils. Compounds common to both EOs were d-elemene, b-elemene, a-bergamotene, b-selinene, caryophyllene oxide and ledene oxide, but the concentration in the oil from the wild population was lower. The hydrocarbon sesquiterpene b-guaiene was found in the wild but not in the cultivated population. On the other hand, the oxygenated sesquiterpenes humulene-1,2-epoxide, isoshyobunone were found only in the cultivated population.
The major fraction of sesquiterpene hydrocarbons in R. carthamoides EO from the wild and cultivated populations were also previously reported in EOs from R. carthamoides hairy roots and roots of soil-grown plant [15]. On the contrary, in EO isolated from R.carthamoides roots and rhizomes grown in Poland monoterpenes were the prevalating fraction [13]. The following volatile compounds—13-norcypera-1(5),11(12-diene) (15.3 and 22.6%), cyperene (17.9% and 18.2%) and cadalene (7.2%)—had previously been reported as the main chemical compounds isolated from R. carthamoides hairy root EO [13,14]. We found that cyperene was one of the major compounds in R. carhamoides EO from the wild population, but this compound was not found in EO from the cultivated population. Humulene was not previously reported, but we found it in both oils, and in the cultivated sample concentration the percentage was higher. Previous studies reported b-elemene in a lower concentration than was reported in the current study. Although b-elemene was identified in both samples, this compound was found in a higher amount in the oil isolated from the cultivated population. The concentration of b-selinene was almost the same when comparing the amounts previously reported and those discovered in the current investigation. Caryophyllene oxide had been found previously, and it was found in both samples in our study. Alpha-pinene, rotundene and limonene had been found in previous studies [13,14], and our performed analyses confirmed their presence only in the EO from the wild population. The following major compounds: cyperene, trans-b-bergamotene, petasitene, 2,5,8-trimethyl-1-naphthol, dauca-4(11),8-diene, nardosina-1(10),11-diene and aplotaxene were found in previously reported chemical composition of EO from R. carthamoides roots of soil-grown plants. In contract, in both of our analyzed EOs cyperene were found only in the sample from the wild grown R. carthamoides. The major compounds in R. carhamoides EO from the wild population estragole, D-carvone, b-selinene and ledene oxide were not previously reported in EO from R. carthamoides roots of soil-grown plants. Moreover, amoung the major chemical compounds in EO from the cultivated population only humulene was previously reported in the oil sample from R. carthamoides roots of soil-grown plants [14].
The results obtained in this study showed differences in chemical composition of the both analyzed EOs. The contents of compounds and their amounts in EOs may vary depending on geographic origin, individual plant chemotypes, time of harvest and habitat conditions—soil, soil management and climate [21–23]. Different chemical compositions of EOs may lead to changes in the biological activity. In this regard strict monitoring of chemical composition is important if different sources of plant material are used. Further studies are needed to investigate the potentially altered pharmacological properties of the EO from R.carthamoides populations and the relationship between chemical composition and pharmacological effects. Moreover, further studies on the cultivation conditions of R. carthamoides in Bulgaria are necessary. This may increase the importance and use of R. carthamoides EO.
- Reviewer 1 comment 25:
- Please specify for which group of readers the presented research will be useful
Authors’ response:
Dear reviewer,
The manuscript is aimed for researchers interested in essential oils, their histochemical analysis and chemical composition. It is also aimed at the scientists involved in the study of R.carthamoides.
- Reviewer 1 comment 26:
- Indicate the research perspectives in the future
Authors’ response:
Dear reviewer, for future studies, it would be good to improve the acculturation conditions to obtain a better quality chemical composition of the oil. Moreover it would be great to be studied the relationship between chemical composition and new bioactivities.
- Reviewer 1 comment 27-29:
Conclusion
- l. 249-253 this information should not be placed here
28 L. 263-265 please re-think the sentence, especially “…the diagnostic purposes”... ??, „…quality control of the plants’ drugs.” ???
- Draw logical and concrete conclusions from the present study
Authors’ response:
Dear reviewer,
Thank you for your recommendations. We have rewrite it.
The histochemical analysis of R. carthamoides roots and rhizomes from wild and cultivated populations confirmed the presence of lipophilic substances in the secretory ducts. The conducted histochemical analysis could assist the pharmacognostic one and in particular the microscopic identification of the tested herbal drug, which could improve the quality control in the presence adulteration. In addition, it gives information on the exact localization of the contained EO as a secondary metabolite. Furthermore, the data obtained from GC-MS analysis indicated that EO from R. carthamoides cultivated population did not contain monoterpene hydrocarbons. In EOs from wild and cultivated population, the sesquiterpene hydrocarbons were the major fraction at 28.23% and 45.10%, respectively. The major compounds in EO from the wild population were b-selinene (4.77%), estragole (6.32%), D-carvone (6.37%), cyperene (8.78%) and ledene oxide (11.52%). The main identified compounds in cultivated population were humulene (7.68%), b-elemene (10.76%), humulene-1,2-epoxide (11.55%), ledene oxide (13.50%) and d-elemene (19.08%).
The knowledge of R. carthamoides cultivated in Bulgaria is enhanced by investigating for the first time the composition of its EO. In view of our results regarding the differences in chemical composition between the EOs from the wild and cultivated populations, further studies on improving the cultivation conditions of R. carthamoides in Bulgaria may be necessary.
- Reviewer 1 comment 30:
References
- Check the list of references (l. 307-309, 317-322 etc.) and the entire text so that they follow the guidelines for authors
Authors’ response:
Dear reviewer,
Thank you for your recommendations. The references are according to mdpi citing stile.

Reviewer 2 Report
The article authored by Todorova and co-workers uses histochemical screening and GC-MS analyses to address the differential accumulation of essential oil components in subterranean organs of wild and cultivated Rhaponticum carthamoides.
The observed difference among wild and cultivated samples is suggested to be due to the variables in habitat/growth conditions, and post harvesting process (lines 29-30, 240-242). Without testing, the article concludes (line 267) that improving the cultivation process is important to have a better essential oil yield and/or pattern. Many plants have shown different chemical compositions upon changing the cultivation conditions. The new aspect would be to spot the conditions that can affect the components of the essential oil.
- The authors know that improvement is a relative word that is always compared with a start point. The article should be improved in a direction that not only helps other laboratories/readers to have a clearer view about the cultivation process, but also how to improve it. How expert readers can benefit from the current finding without knowing the cultivation process used in this article? It could be helpful if the authors clearly describe the climate (ex. soil, temperature range, geographic coordinates) and the post-harvesting process including the storage for wild and cultivated plants studied in the current article. If it seems hard to precisely define the post- and cultivation conditions of the wild pant, at least the estimation should be written accompanied by a clear description for the cultivated plant. The authors should pick some cultivation conditions to be tested for possible improvements.
-An additional figure showing possible morphological differences between wild and cultivated samples should be added (including scale bars).
-Subsection 2.2: the authors should clearly describe the climate and growth conditions of the cultivated plant. This should include soil type, temperature range, day hours. The harvesting time should also be mentioned for wild and cultivated samples.
-Subsection 2.4: description of the drying process is missing. Which solvent was used to collect the EO? Please, add its volume.
-Subsection 3.1 (line 131) the sections do not show clearly if the secretory ducts are surrounded with one or more layers of secretory epithelial cells. It would be better to show a photo with a single layer as to match the described test.
-Lines 131-132: how the authors know that these cells are for synthesis and secretion? To my knowledge, there is no localization of the enzymes responsible for the biosynthesis have been confirmed in these organs so far. I agree that it is a place for secretion, but synthesis needs further experiments. If transporters are involved, the place of synthesis might be different.
-Figures 1 and 2: Please, carefully consider to properly edit all scale bars. The numbers should be visible, and the scale bar must be complete.
-Figures 3 and 4: Please write the full name of GCPs; the Y-axis label. Please, italicize the plant name in the legends. Also, re-edit these two figures to highlight the chromatogram with zones according to the length of the carbon chain, referring to the used standard alkanes mix.
-Line 112: please type the numbers in C8-C30 to be subscript as they present the carbon chain length, not a numbering of an exact carbon.
-Table 1: The legend shows that tr is detection less than 0.05%. Please add that '-' means not detected, and describe what is 't'.
-The symbol used before each compound should be well-accepted (ex. α, β, ѵ,…etc). Please, revise the used symbol before pinene, linalool, terpineol , copaene, etc.
Author Response
Dear reviewer,
Thank you for your remarks and recommendations.
Reviewer 2 comments:
- Reviewer 2 comment 1:
- The authors know that improvement is a relative word that is always compared with a start point. The article should be improved in a direction that not only helps other laboratories/readers to have a clearer view about the cultivation process, but also how to improve it. How expert readers can benefit from the current finding without knowing the cultivation process used in this article? It could be helpful if the authors clearly describe the climate (ex. soil, temperature range, geographic coordinates) and the post-harvesting process including the storage for wild and cultivated plants studied in the current article. If it seems hard to precisely define the post- and cultivation conditions of the wild pant, at least the estimation should be written accompanied by a clear description for the cultivated plant. The authors should pick some cultivation conditions to be tested for possible improvements.
Authors’ response:
Dear reviewer, Thank you for your recommendations. The purpose of the study is not to investigate the accumulation of volatile substances during the different stages of ontogenesis, this would be the subject of future research.
- Reviewer 2 comment 2:
-An additional figure showing possible morphological differences between wild and cultivated samples should be added (including scale bars).
Authors’ response:
Dear reviewer, Thank you for your recommendations. The study concerned the histo-anatomical characteristics and localization of essential oil in the samples of wild and cultivated R. carthamoides roots and rhizomes.
- Reviewer 2 comment 3:
-Subsection 2.2: the authors should clearly describe the climate and growth conditions of the cultivated plant. This should include soil type, temperature range, day hours. The harvesting time should also be mentioned for wild and cultivated samples.
Authors’ response:
Dear reviewer,
Thank you for your recommendations.We do not have information about growth conditions of the cultivated R. carthamoides, because we have purchased the plant material
- Reviewer 2 comment 4:
-Subsection 2.4: description of the drying process is missing. Which solvent was used to collect the EO? Please, add its volume.
Authors’ response:
Dear reviewer,
Thank you for your recommendations. We have not used a solvent to collect the essential oil. Removal of water droplets was accomplished by adding a small amount of anhydrous sodium sulfate (drying process)
- Reviewer 2 comment 5:
-Subsection 3.1 (line 131) the sections do not show clearly if the secretory ducts are surrounded with one or more layers of secretory epithelial cells. It would be better to show a photo with a single layer as to match the described test.
Authors’ response:
Dear reviewer,
Thank you for your recommendations. The text had been changed.
Microscopic observations revealed the presence of internal secretory ducts filled with orange-brown secretion in the roots and rhizomes of the examined wild and cultivated samples of R. carthamoides (Figure 1 and Figure 2).
A characteristic feature in the root transverse sections of both investigated populations was the arrangement of the secretory ducts in the form of a ring along the endoderm between the parenchyma of the primary cortex and the secondary phloem (Figure 1A and Figure 2A).
In the rhizome transverse section of the wild R. carthamoides the secretory ducts were scattered throughout the cortical parenchyma and the secondary phloem (Figure 1D), while in the rhizome of the cultivated population these oil secretory structures were observed as characteristic arches arranged on a group of lignified fibers, capping the vascular bundles (Figure 2D). Such distinctions can be attributed to the different stages of ontogenesis.
However, the secretory system structures in the examined R. carthamoides roots and rhizomes were similar. The secretory ducts consisted of intercellular spaces surrounded by specialized epithelial cells responsible for the volatile oil secretion (Figure 1B, E and Figure 2B, E).
The lipid nature of the duct content and related surrounding epithelial cells was confirmed by orange-red staining with Sudan III (Figure 1C, F and Figure 2C, F). No differences were detected between wild and cultivated samples of the R. carthamoides root and rhizome from a histochemical perspective.
The results of the performed microscopic histochemical analysis were in accordance with the data specified in the Russian Pharmacopoeia [4] and also corresponded to the anatomical studies on the vegetative organs of R. carthamoides [3].
- Reviewer 2 comment 6:
-Lines 131-132: how the authors know that these cells are for synthesis and secretion? To my knowledge, there is no localization of the enzymes responsible for the biosynthesis have been confirmed in these organs so far. I agree that it is a place for secretion, but synthesis needs further experiments. If transporters are involved, the place of synthesis might be different.
Authors’ response:
Dear reviewer,
Thank you for your recommendations. The text had been changed.
- Reviewer 2 comment 7:
-Figures 1 and 2: Please, carefully consider to properly edit all scale bars. The numbers should be visible, and the scale bar must be complete.
Authors’ response:
Dear reviewer,
Thank you for your recommendations. We have increased the scale bar on the figures and added them individually for each picture to the legend.
- Reviewer 2 comment 8:
-Figures 3 and 4: Please write the full name of GCPs; the Y-axis label. Please, italicize the plant name in the legends. Also, re-edit these two figures to highlight the chromatogram with zones according to the length of the carbon chain, referring to the used standard alkanes mix.
Authors’ response:
Dear reviewer,
Thank you for your recommendations. We added the missing information in Figures description. Following the recommendations of another reviewer, we numbered the main peaks of the chromatograms.
Reviewer 2 comment 9:
-Line 112: please type the numbers in C8-C30 to be subscript as they present the carbon chain length, not a numbering of an exact carbon.
Authors’ response:
Dear reviewer,
Thank you for your recommendations. We have changed it.
- Reviewer 2 comment 10:
-Table 1: The legend shows that tr is detection less than 0.05%. Please add that '-' means not detected, and describe what is 't'.
Authors’ response:
Dear reviewer, Thank you for your recommendations. 't' actually was the technical mistake, the correct was tr, we have changed it. We added the mean of “-” in description.
- Reviewer 2 comment 11:
-The symbol used before each compound should be well-accepted (ex. α, β, ѵ,…etc). Please, revise the used symbol before pinene, linalool, terpineol , copaene, etc.
Authors’ response:
Dear reviewer,
Thank you for your recommendations. This was a technical mistake. We have corrected it.

Reviewer 3 Report
Please see the attached pdf file.

Author Response
Dear reviewer,
Thank you for your recommendations.
Reviewer 3 comments:
- Reviewer 3 comment 1:
- Line 112- italic the n-alkanes
Authors’ response:
Dear reviewer,
Thank you for your recommendations. We have changed it.
- Reviewer 3 comment 2:
- Scale bars in Figure 1 is not clear
Authors’ response:
Dear reviewer,
Thank you for your recommendations. We have made it clear.
- Reviewer 3 comment 3:
- Italic the R. carthamoides in figure 1 and 2
Authors’ response:
Dear reviewer,
Thank you for your recommendations. It was technical mistake, we have corrected it.
- Reviewer 3 comment 4:
- Line 159 ;.
Authors’ response:
Dear reviewer, Thank you for your recommendations. We have corrected it.
- Reviewer 3comment 5:
- Figure 3 what is the major peak
Authors’ response:
Dear reviewer, Thank you for your recommendations. We have numbered it.
- Reviewer 3 comment 6:
- Figure 3 and 4 - I suggest that authors add some major or important peak numbers in the chromatograms.
Authors’ response:
Dear reviewer,
Thank you for your recommendations. We have numbered it.
- Reviewer 3 comment 7:
- Figure 3 and 4- italic R. carthamoides
Authors’ response:
Dear reviewer,
Thank you for your recommendations. We have corrected it.
- Reviewer 2 comment 8:
- I suggest that the major components of EOs should identify by using co-injection of authentic compounds.
Authors’ response:
Dear reviewer,
Thank you for your recommendations. Only qualitative analysis was done, future studies will be aimed at full identification with reference samples, quantitative analysis and relationship between the chemical composition and pharmacological effects of the essential oil.
- Reviewer 2 comment 9:
- Strange simbols in table 1
Authors’ response:
Dear reviewer,
Thank you for your recommendations. It was technical mistake, we have corrected it.
- Reviewer 3 comment 10:
- RIs of these 3 elemenes did not match with the literature data of Adams (2007). Please check.
Authors’ response:
Dear reviewer,
Thank you for your recommendations. We have used also and NIST. It was technical mistake, we have corrected it.
- Reviewer 3 comment 11:
- The authors mentioned in the manuscript that no significant difference was observed from wild and cultivated species. How does "histochemical localization" may support the diagnostic purposes and quality control of the plants’ drugs ?
Authors’ response:
Dear reviewer,
Thank you for your recommendations. We have rewrite the data.
The histochemical analysis of R. carthamoides roots and rhizomes from wild and cultivated populations confirmed the presence of lipophilic substances in the secretory ducts. The conducted histochemical analysis could assist the pharmacognostic one and in particular the microscopic identification of the tested herbal drug, which could improve the quality control in the presence adulteration. In addition, it gives information on the exact localization of the contained EO as a secondary metabolite. Furthermore, the data obtained from GC-MS analysis indicated that EO from R. carthamoides cultivated population did not contain monoterpene hydrocarbons. In EOs from wild and cultivated population, the sesquiterpene hydrocarbons were the major fraction at 28.23% and 45.10%, respectively. The major compounds in EO from the wild population were b-selinene (4.77%), estragole (6.32%), D-carvone (6.37%), cyperene (8.78%) and ledene oxide (11.52%). The main identified compounds in cultivated population were humulene (7.68%), b-elemene (10.76%), humulene-1,2-epoxide (11.55%), ledene oxide (13.50%) and d-elemene (19.08%).
The knowledge of R. carthamoides cultivated in Bulgaria is enhanced by investigating for the first time the composition of its EO. In view of our results regarding the differences in chemical composition between the EOs from the wild and cultivated populations, further studies on improving the cultivation conditions of R. carthamoides in Bulgaria may be necessary.

Round 2
Reviewer 1 Report
The article may be further processed for printing.
Congratulations to the Authors.
Please remove duplicate phortography - figure 1A-F, Figure 2A-F.
Kind regards
Author Response
Reviewer 1 comments:
The article may be further processed for printing.
Congratulations to the Authors.
Please remove duplicate phortography - figure 1A-F, Figure 2A-F.
Kind regards
Authors’ response:
Dear reviewer,
Thank you for your review and recommendations. We have removed the duplicates of the Figures 1 and 2.
Reviewer 2 Report
The article does not have any changes related to comment 1, 2 and 3, which, in my view, affects the reproducibility and soundness of this work.
Author Response
Reviewer 2 comments:
The article does not have any changes related to comment 1, 2 and 3, which, in my view, affects the reproducibility and soundness of this work.
Authors’ response:
Dear reviewer,
Thank you for your review and recommendations.
- Reviewer 2 comment 1:
- The authors know that improvement is a relative word that is always compared with a start point. The article should be improved in a direction that not only helps other laboratories/readers to have a clearer view about the cultivation process, but also how to improve it. How expert readers can benefit from the current finding without knowing the cultivation process used in this article? It could be helpful if the authors clearly describe the climate (ex. soil, temperature range, geographic coordinates) and the post-harvesting process including the storage for wild and cultivated plants studied in the current article. If it seems hard to precisely define the post- and cultivation conditions of the wild pant, at least the estimation should be written accompanied by a clear description for the cultivated plant. The authors should pick some cultivation conditions to be tested for possible improvements.
Authors’ response:
Dear reviewer, Thank you for your recommendation. The aim of our study was to investigate and compare the chemical composition and histochemical localization of essential oils isolated form R. carthamoides species wild-grown and cultivated in Bulgaria. We understand that the different stages of ontogenesis and the post-harvest process are important to chemical composition of essential oils, but it’s not the main aim of our study. For further studies we will examined the cultivation process, harvest time and their influence on chemical composition of the essential oil, but the main aim now of this study is to compare the chemical composition of R. carthamoides species grown in different geographic regions.
- Reviewer 2 comment 2:
-An additional figure showing possible morphological differences between wild and cultivated samples should be added (including scale bars).
Authors’ response:
Dear reviewer, Thank you for your recommendation. However, the data about the morphological analysis of R. carthamoides is included in Russian Pharmacopoeia. The morphological analysis was used only for identification of the two samples. We did not find differences between the examined samples, so we decided not to include morphological analysis and to show only histochemical localization of essential oil, because there is no previous data about it and our aim is for the essential oil isolated from R.carthamoides.
- Reviewer 2 comment 3:
-Subsection 2.2: the authors should clearly describe the climate and growth conditions of the cultivated plant. This should include soil type, temperature range, day hours. The harvesting time should also be mentioned for wild and cultivated samples.
Authors’ response:
Dear reviewer,
Thank you for your recommendation. We have added the following data about cultivated sample in subsection 2.1. Plant materials: The cultivated plant has grown in a continental climate with an average annual temperature 11.9oC and diluvial meadow cinnamon soil.